

**Black Carbon Seasonal and Diurnal Variation in surface snow in Svalbard and its**
**Connections to Atmospheric Variables**
Michele Bertò[1#], David Cappelletti[2,7], Elena Barbaro[1,3], Cristiano Varin[1], Jean-Charles Gallet[4],
Krzysztof Markowicz[5], Anna Rozwadowska[6], Mauro Mazzola[7], Stefano Crocchianti[2], Luisa
Poto[1,3], Paolo Laj[8], Carlo Barbante[1,3] and Andrea Spolaor[1,2].
[1]Ca' Foscari University of Venice, Dept. Environmental Sciences, Informatics and Statistics, via Torino, 155 -
30172 Venice-Mestre, Italy;
[2]Università degli Studi di Perugia, Dipartimento di Chimica, Biologia e Biotecnologie, Perugia, Italy;
[3]CNR-ISP, Institute of Polar Science – National Research Council –via Torino, 155 - 30172 Venice-Mestre, Italy;
[4]Norwegian Polar Institute, Tromsø, Norway.
[5]University of Warsaw, Institute of Geophysics, Warsaw, Poland;
[6]Institute of Oceanology, Polish Academy of Sciences, Sopot, Poland;
[7]CNR-ISP, Institute of Polar Science – National Research Council – Via Gobetti 101, Bologna;
[8]Univ. Grenoble-Alpes, CNRS, IRD, Grenoble-INP, IGE, 38000 Grenoble, France
[#] Now at Laboratory of Atmospheric Chemistry, Paul Scherrer Institute, 5232 Villigen PSI, Switzerland
**Abstract**
Black Carbon (BC) is a major forcing agent in the Arctic but substantial uncertainty remains to
quantify its climate effects due to the complexity of mechanisms involved. In this study, we provide
unique information on processes driving the variability of BC mass concentration in surface snow in the
Arctic. Two different snow-sampling strategies were adopted during spring 2014 and 2015, focusing on
the *refractory* BC (rBC) mass Ny-Ålesund concentration daily/hourly variability on a seasonal/daily time
scale (referred to as 80-days and 3-days experiments). Despite the low rBC mass concentrations (never
exceeding 22 ng g$^{-1}$), a daily variability of up to 4.5 ng g$^{-1}$ was observed. Atmospheric, meteorological and
snow-related physico-chemical parameters were considered in multiple statistical models to understand
the factors behind the observed variation of rBC mass concentrations. Results indicate that the main
drivers of the variation of rBC are the precipitations events, snow metamorphism (melting-refreezing
cycles, surface hoar formation and sublimation) and the activation of local sources (wind resuspension)
during the snow melting periods. The rBC in the snow seems de-coupled with the atmospheric BC load.
Our results highlighted a common association of snow rBC with coarse mode particles number
concentration and with snow precipitation events.





## 1. Introduction


In the last two decades, the Arctic regions have been exposed to dramatic changes in terms of
atmospheric temperature rise, sea ice decreasing and increase of air mass transport from lower latitudes
bringing warmer and humid air masses containing pollutants and anthropogenic derived compounds (Law
and Stohl, 2007; Comiso et al., 2008; Screen and Simmonds, 2010; Eckhardt et al., 2013; Schmale et al.,
2018; Maturilli et al., 2019). Despite a general decreasing trend observed in most parts of the globe, in the
Arctic region the coefficients describing the aerosol optical properties, e.g. scattering and absorption, had
generally   shown a weakly increasing to a not-statistically significant trend (95% confidence),
respectively (Collaud Coen et al., 2020). Long-range transport and local emissions of combustion
generating aerosols as black carbon (BC) could influence the radiative budget of the Arctic atmosphere,
especially after the impacts of atmospheric aging on the mixing state of BC particles (Eleftheriadis et al.,
2009; Bond et al., 2013; Zanatta et al., 2018). When deposited over snow, many aerosol species directly
absorb the solar radiation more efficiently than snow itself, thus favoring snow aging processes and the
decrease of the snow albedo (Hansen and Nazarenko, 2004; Flanner et al., 2007; Hadley and Kirchstetter,
2012; Skiles et al., 2018; Skiles and Painter, 2019).
Among these light-absorbing aerosols, *black carbon* (BC) particles are the most effective in
absorbing the visible and near infrared solar radiation. These primarily emitted, insoluble, refractory and
carbonaceous particles originate from natural and anthropogenic sources such as open fires or diesel
engine exhausts, respectively. Currently, the anthropogenic emissions are higher compared to the natural
ones (Moosmüller et al., 2009; Bond et al., 2013). In 2000, approximately 59% of the total global BC
emissions were generated from the energy production sector (including fossil fuels and solid residential
fuels combustions) and the remaining from biomass burning (Bond et al., 2013). BC particles account for
about 10% of the total aerosol mass in the European atmosphere and are characterized by a mass size
distribution peaking  around 100-250 nm (of mass equivalent diameter), e.g. at 240 nm in the Svalbard
area in spring  (Bond et al., 2013; Laborde et al., 2013; Zanatta et al., 2016; Motos et al., 2019). In a
global perspective the BC radiative forcing (RF) is considered to be second only to that of $CO_2$, even
though the value is characterized by a 90% uncertainty (Bond et al., 2013). The impact of BC particles
absorbing the incoming solar radiation has certainly a non-negligible role in the Arctic region which is
already threatened by a two-fold temperature increase compared to the mid-latitude regions, the so called
"Arctic Amplification" (Bond et al., 2013; Cohen et al., 2014; Serreze and Barry, 2011). BC has an
atmospheric lifetime of about seven days and has been directly targeted in important international
mitigation agreements (Programme (AMAP), 2015). In this manuscript,  the recommended nomenclature



proposed in Petzold et al. (2013) is used to described the BC related quantities, depending on the
deployed instrumental method.
Theoretical and experimental results showed that the cryosphere is affected both by the BC-
induced warming of the atmosphere and by direct and indirect BC effects on the snow once deposited
over it (Flanner, 2013), as for example the increase of absorbed incoming solar radiation by BC at snow
surface. Consequently, snow is melting faster, decreasing the snow cover period, but snow is also aging
more rapidly, which further decreases the snow albedo (Ramanathan and Carmichael, 2008; Brandt et al.,
2011; Hadley and Kirchstetter, 2012).
Atmospheric BC measurements in the Arctic regions are still rare, despite an extraordinary effort
done by the international scientific community in order to evaluate the sources, transport paths,
concentration and climate impact  (Eleftheriadis et al., 2009; Pedersen et al., 2015; Ferrero et al., 2016;
Ruppel et al., 2017; Osmont et al., 2018; Zanatta et al., 2018; Laj et al., 2020). In order to understand the
behavior of the BC particles in the snow mantle and to retrieve their radiative impact, several studies were
performed in the last decades, measuring the amount of BC particles and their properties directly in the
snow. A detailed list of previous measurements of BC in the Arctic snow pack (using different
methodologies) is reported in Forsström et al. (2009) focused on snow surface in Svalbard and measured
the *elemental carbon* (EC) content using filters and thermo/optical method. The results from 81 samples
indicate a large variability in the EC mass concentration, ranges from 0 to about 80 ng g$^{-1}$ (with a median
mass concentration of about 4 ng g$^{-1}$). In Aamaas et al. (2011), the EC carbon mass concentration was
measured in surface snow samples collected around the settlements of Longyearbyen and Svea in the
Svalbard archipelago, that are affected by intense local sources (i.e. diesel, coal power plants and coal
extraction). There, EC mass concentration in the snow samples was up to 1000 ng g$^{-1}$, rapidly decreasing
with distance (50 ng g$^{-1}$ at a distance of about 5 km). Oppositely, the snow collected near Ny-Ålesund was
characterized by an average EC concentration of 6.6 ng g$^{-1}$ with a standard deviation of 4.3 ng g$^{-1}$,
defined as an Arctic background-like concentration. Moreover, Aamaas et al. (2011) observed that if the
surface snow is influenced by melting episodes most of the BC-containing particles remain on the
surface, therefore virtually increasing their mass concentration, especially during the spring season. Snow
samples from Scandinavia and European Arctic were analyzed and discussed in Forsström et al. (2013) in
terms of EC content: in the Scandinavian snow samples the EC concentrations were up to ~47 ng g$^{-1}$, due
to local emissions, whereas in more remote regions as in Barrow (Northern Alaska) they were from 3 to
14 ng g$^{-1}$. The data discussed in Pedersen et al. (2015) from Arctic snow (Ny-Ålesund, Tromsø, Fram
Strait and Barrow), also measured with the thermo-optical method were characterized by an EC
concentration ranging from 5 to 137 ng g$^{-1}$. Specifically, the surface snow samples from Ny-Ålesund in



2010 and 2011 had a median EC concentration from 18 to 21 ng g$^{-1}$. In Gogoi et al. (2016) snow samples
were collected in the surrounding area of the atmospheric BC observatory at Gruvebadet (Ny-Ålesund,
Svalbard) during April 2012. They used a filter based absorption method (Clarke and Noone, 1985), i.e. a
dual wavelength optical transmissometer (Sootscan, Model OT-21, Magee Scientific, USA) at 880 and
370 nm. The *equivalent* BC (eqBC; derived from absorption measurements) mass concentrations in their
samples ranged from 0.6 to 4.1 ng g$^{-1}$, and the fraction of BC from biomass burning was up to 25%. Khan
et al. (2017) selected two sites, Woodfjorden and a coal dust contaminated site in southern Svalbard
(Mine 7), to collect background-like and coal dust affected surface snow samples, respectively.
Concentrations of EC varied from 5 to about 4000 ng g$^{-1}$. Also the refractory black carbon (rBC, i.e.
measured with a Single Particle Soot Photometer – SP2, DMT) concentration values were reported
ranging from 1 to 340 ng g$^{-1}$. The difference between the values obtained using the thermo-optical and the
laser-induced incandesce (SP2) methods arises from the physical principles involved in the measurement,
from the different size ranges and from differences in the aerosols physico-chemical characteristics. A
review of the various methodologies to measure BC can be found in Bond et al. (2013), whereas a
nomenclature definition in Petzold et al. (2013). Mori et al. (2019) analyzed the rBC mass concentration
and size distribution for snow samples from several regions in the Arctic (Greenland, Finland, Alaska,
Siberia, and Svalbard) showing a latitudinal variability, consistent with changes in anthropogenic BC
emissions, atmospheric precipitable water content and topography changes.

A complex combination of processes are involved in the BC particles transfer from the

atmosphere to the surface snow. The wet deposition is traditionally considered as the main scavenging
process, particularly efficient for particles in the typical atmospheric BC size range. For bigger particles,
instead, dry deposition is generally more efficient. Via a modelling approach, Liu et al. (2011) found that
approximately 50% of the total burden of BC in the Arctic atmosphere is removed through wet deposition
related processes. Yasunari et al. (2013) estimated the intensity of BC dry deposition on the Himalayan
glaciers using several dry deposition methods (models and observations). Particularly, they found that the
surface roughness and the surface wind speed are critical parameters in order to retrieve realistic results.
Emerson et al. (2018) empirically evaluated the in situ rBC deposition velocities over a grassland (0.3 ±
0.2 mm s$^{-1}$), suggesting eddy covariance as the main deposition driver. In a recent study, Jacobi et al.
(2019) confirmed the previous estimates of the importance of the wet deposition in removing BC particles
in the atmosphere. Their results suggest that in spring and in the Svalbard Arctic area, approximately 60%
of the BC particles are deposited on the surface snow via wet deposition. Moreover, they found out that
the BC particles deposition is similar to those of nitrate and non-sea-salt (nss) sulfate, equally explained
through wet and dry deposition (in contrast with the major sea salt components, mainly deposited via wet
deposition).


Complex air-snow BC transfer, post depositional processes and potentially high radiative impacts
make the BC behavior in the Arctic snow pack an intriguing and complex research topic. BC content in
the surface snow is still poorly characterized in the Arctic region, particularly for what concerns
measurements performed with single-particle accurate instruments, i.e. the SP2. The absolute values of
rBC mass concentration are important to evaluate the BC radiative impact via snow albedo reduction. In
this work, the variability of the snow absolute rBC mass concentration was investigated for the first time
following two different sampling frequencies, daily and hourly. To do this, two field campaigns were
performed in the vicinity of the Gruvebadet Aerosol Laboratory, in Ny-Ålesund, during spring 2014 and
2015. The daily sampling lasted for approximately 80 days, allowing to evaluate the seasonal variability
of BC. The daily sampling covered the transition from a cold period characterized by exceptionally
frequent snow precipitation events to the melting period in late May, characterized by snow surface
melting episodes and the presence of re-suspended surface material consequent to the collapse of the
snow pack. In order to investigate the processes having a non-negligible role in regulating the surface
snow rBC mass concentration, several parameters were deployed in a multilinear statistical model trying
to explain the observed BC variability in the surface snow. Specifically, the statistical model took into
account the atmospheric *equivalent* BC mass concentration, selected meteorological parameters, the snow
coarse mode particles content and chemical parameters.

**2. Experimental Methods**
**2.1 Study Area**

Both experiments were conducted in the proximity of the Ny-Ålesund research station (78.5526
N, 11.5519 E, 25 m a.s.l.) located on the Spitzbergen island in the Svalbard Archipelago. Along the west
coast, Svalbard is characterized by maritime climate with an annual average temperature of -3.9°C in Ny-
Ålesund (between 1994 and 2017), and during that period, temperature increased by 1.6°C/decade
(Maturilli et al., 2019). The measured mean annual temperatures for the years of the two campaigns
described in this work were of -2.76°C (2014) and -2.16°C (2015). The average annual precipitation in
Svalbard ranges from 190 to 525 mm (385 mm in Ny-Ålesund) with the highest precipitation rates
occurring in August-October (mainly rain) and March, while May-June correspond to the lowest rates
(Førland et al. 2011). During winter, snow covers most of the places and is the main interface influencing
the ecosystem and climate system (Hansen et al. 2014). On average, the snow pack starts building up in
September and melts away at the end of May (Førland et al. 2011). As reported in Maturilli et al. (2019),
the year 2014 was characterized by the longest snow cover period, due to the exceptionally intense snow
precipitation events.



Snow samples were collected in the area close to the atmospheric research station of Gruvebadet
(78.91734 N, 11.89535 E, 40 m a.s.l.), about 1 km South-West of Ny-Ålesund (Figure ). Ny-Ålesund has
become one of the reference locations for conducting Arctic climate studies focusing on atmospheric
composition and physics, oceanography, biology, permafrost and snow-related activities as well as for
evaluating the human impact in the higher Arctic. Long-term monitoring of atmospheric aerosols is
performed at the Gruvebadet station (Feltracco et al., 2019; Moroni et al., 2018; Ferrero et al., 2016;
Bazzano et al., 2015; Moroni et al., 2015; Zangrando et al., 2013; Scalabrin et al., 2012) as well as the
Zeppelin observatory (475 m a.s.l.) (Eleftheriadis et al., 2009; Tunved et al., 2013; Lupi et al., 2016, and
reference therein).

**2.2 Snow Sampling**

Snow samples were collected during two field campaigns: in spring 2014, from April 1 to June 24
(85 days in total, daily sampling, referred as "80-days") and in spring 2015 from April 28 to May 1 (three
days, hourly sampling, referred as "3-days"). In the following, the two campaigns will be referred to as
the 80-days and the 3-days experiment, respectively.
Two different sampling schemes were adopted regarding the thickness of the surface snow
samples and the temporal sampling frequency. In the 80-days experiment, the first 10 cm of surface snow
were collected on a daily basis (approximately at 11.00 am, GMT+2) in the same area, using a 5 cm
diameter and 10 cm long Teflon tube. The snow samples were collected leaving a distance of
approximately 15 cm between the previous sample location and following a straight line, in order to
minimize the spatial variability influence. The collected snow was homogenized in a pre-cleaned plastic
bag and then, without melting, a snow aliquot was transferred into a 50 mL vial (Falcon™ 50mL Conical
Centrifuge Tubes) for BC, coarse mode particles number concentration and electrical conductivity
analyses. On April 5, due to a snowstorm, the daily sample was not collected. During the 3-days
experiment, the first 3 cm of surface snow were collected on an hourly basis in pre-cleaned vials in a
delimited area of  2 m x 2 (Spolaor et al., 2019). To minimize the spatial variability, the samples have
been collected following a straight line leaving about 5 cm between the sampling points. The samples for
both experiments were kept frozen until the analyses period. The samples were collected using neck nylon
gloves with particular attention to avoid any contamination from the not covered part and always
downwind of the sampling area.
The temperature of the surface of the snow pack (at 7 cm for 80-days and at 3 cm for 3-days
experiment) was always measured during the sampling with the same resolution. The daily/hourly snow
accumulation was measured by using 4 poles placed around the sampling area as references. Temperature



and accumulation measurements are ancillary data for evaluating snow deposition and ablation
(precipitation/wind/melting).

**2.3 Atmospheric Optical Measurements**
**2.3.1 Aethalometer (AE-31)**

In this study, the equivalent BC (eBC) concentration in the lower atmosphere (around 3 m a.g.l)

was measured by an AE-31 aethalometer (Gundel et al., 1983), during the 3-days campaign. The device is
equipped with 7-wavelengths (370, 470, 520, 590, 660, 880, 950 nm) and determines the attenuation
coefficient by using the ratio of light attenuated through a sensing spot (where aerosols are deposited) and
a referenced clean spot, both on a quartz fiber filter substrate. The sampling and reference spots surface
areas are 0.5 cm$^2$, while volumetric flow rate is 4 l min$^{-1}$. The flow rate was calibrated with a TetraCal
(BGI Instruments) volumetric airflow before and after the field campaign. A 5 minutes temporal
resolution was used for data acquisition. However, due to the low background concentration in the Arctic,
the signal/noise ratio is high, so that data were average in an hour interval. Most of the filter-based
techniques used to measure the aerosol absorption coefficient and eBC suffer from different systematic
errors that must be corrected. In case of the aerosol absorption coefficient, the most important are the
corrections for multiple scattering by the filter fibers and aerosol particles, and for filter loading effects.
The data presented in this study were processed according to Segura et al. (2014) methodology. For this
purpose the multiple scattering and filter loading effect (Weingartner et al., 2003) was corrected with new
values of mass absorption cross section (MAC) and multiple scattering factor (C=3.1) reported by Zanatta
et al. (2018). The MAC value was derived using observations and observationally constrained Mie
calculations in spring at the Zeppelin Arctic station (Svalbard, 78°N). Zanatta et al. (2018) estimated the
MAC at 550 nm (9.8 m2 g-1) and at 880 nm (6.95 m2 g-1), which we used to estimated MAC at 520 nm
(10.2 m2 g-1).

**2.3.2 Particle Soot Absorption Photometer (PSAP)**

During the 80-days sampling period the aerosol absorption coefficient was also measured by

means of a 3-wavelengths PSAP. It measures the variation of the transmission of light through a filter
where particles are continuously deposited by a constant airflow. A second filter identical to the first one
remains clean and is used as reference to take into account possible variations of the light source, i.e. a 3-
color LED (blue, green and red with wavelength centered around 470, 530 and 670 nm, respectively). The
correction developed by Bond et al. (1999) was applied to take into account the filter loading effect. The
complete eBC mass concentration time series for the 80-days experiment was retrieved using the
Aethalometer (first period) and the PSAP (second period), with an overlapping period with simultaneous



measurements of 5 days. In order for the retrieved eBC mass concentration from the two instruments to be
equal during the overlapping period, the PSAP eBC was calculated dividing the absorption measurements
(at 530 nm) with a MAC equal to 7.25 $m^2$ $g^{-1}$ (keeping the AE31 data as reference). From the 1-minute
data, daily averages were calculated to compare with the rBC daily data obtained from the snow.

**2.4 Surface Snow measurements**
**2.4.1 Coarse Mode Particles Number Concentration**
The snow samples were melted at room temperature before the on-line coarse-mode particles and
conductivity measurements (the water was pumped from the vials by a 12 channels peristaltic pump,
ISMATECH, type ISM942). Specifically, the number concentration of coarse mode particles in the
surface snow was measured with a Klotz Abakus laser sensor particles counter. This instrument optically
counts the total number of particles and measures the size of each particles in a liquid constantly flowing
through a laser beam cavity (LDS 23/23). The size range of this instrument is from 0.8 to about 80 µm
with 32 dimensional bins (Table SI 1), not overlapping with that of the SP2. Only the 32nd bin has a
dimensional range above 15.5 µm, i.e. of 80 µm. The data were recorded by a LabView® based software
obtaining a sufficient number of data points in order to have a standard deviation of the mean smaller than
5%. The particles number concentration was calculated using the constant water flow value.

**2.4.2 rBC Measurement – SP2**
The rBC mass concentration and mass size distribution were measured following the methods
described in Lim et al. (2014). Particularly, the snow samples were melted at room temperature prior to
the analyses. The vials with the melted snow water were sonicated for ten minutes in water at room
temperature. A glass nebulizer was used with filtered compressed air to nebulize the sample before the
injection in the Apex-Q desolvation system (APEX-Q, Elemental Scientific Inc., Omaha, USA). The
nebulization efficiency was evaluated daily by injecting Aquadag® solutions with different mass
concentrations, ranging from 0.1 to 100 ng $g^{-1}$, obtaining an average value of 61%, that was used to
correct all the BC mass concentrations reported in this manuscript. More details on the method can be
found in Lim et al. (2014) and in Wendl et al. (2014).
A complete description of the theory of the SP2 functioning can be found in Stephens et al.
(2003) and in Moteki and Kondo (2007) and Moteki and Kondo (2010). Briefly, the SP2 measurements
are based on the laser-induced incandescence of the BC particles flowing through a high energy Nd-YaG
laser with a wavelength of 1054 nm, at a single-particle level. The BC particles vaporize at about 4000 K
emitting an incandescence signal proportional to their mass. The SP2 empirical calibration was performed
using the standard reference fullerene soot (obtained from Alfa Aesar, stock #40971 and lot #FS12S011;



the same used during the SP2 inter-comparison described in Laborde et al. (2012); Gysel et al. (2011)).
During the calibration, the fullerene soot particles were size selected in terms of mobility diameter with a
differential mobility analyzer (DMA), ranging from 80 to 500 nm. The calibration points were fitted using
a linear fit. The mass equivalent diameter is calculated assuming the sphericity of the BC particles and an
effective density of 1.8 g cm$^{-3}$ (Moteki and Kondo, 2010).
The SP2 data were analyzed using the IGOR based toolkit from M. Gysel (Laboratory of
Atmospheric Chemistry, Paul Scherrer Institute, Switzerland). The large amount of signals derived from
every single particle are elaborated achieving rBC mass and number concentrations and size distributions.
It's important to remark that the eBC and the rBC mass concentrations are not exactly the same physical
quantities: the former is obtained from an absorption measurement assuming a constant MAC, whereas
the second is obtained via a laser-induced-incandescence method with an SP2 empirically calibrated with
a reference material (Petzold et al., 2013). Given the lack of a detailed mixing state characterization of the
BC particles during the two experiments, it was not possible to predict or estimate the uncertainty
introduced in the statistical analyses resulting from the two different measuring techniques. However, the
statistical analyses are only related to the time variability of these two quantities and not to their absolute
values.

### 284 2.4.3 Conductivity and sodium/manganese concentrations

The total conductivity of the melted snow was measured in parallel with a simple conductivity
Micro-Cell. The water conductivity depends from the amount of soluble anions and cations in the snow,
as for instance sea salt sodium. Concentrations of sodium (Na) and manganese (Mn) were also determined
as tracer of sea spray emission and dust deposition by Inductively Coupled Plasma Sector Field Mass
Spectrometry (ICP-SFMS; Element2, ThermoFischer, Bremen, Germany) equipped with a cyclonic
Peltier-cooled spray chamber (ESI, Omaha, USA). The sample flow was maintained at 0.4 mL min$^{-1}$.
Detection limits, calculated as three times the standard deviation of the blank, were 0.5 ng g$^{-1}$ for $^{23}$Na and
0.3 ng g$^{-1}$ for Mn. The residual standard deviation (RSD) for Na and Mn ranged between 2–5%.

### 294 2.5 Meteorological Parameters

Several meteorological parameters have been used in the statistical exercise to relate the snow
samples to the atmospheric conditions. Air temperature and relative humidity at 2 meter height have been
retrieved from a meteorological station located about 800 meters north of the sampling site, using a
ventilated PT-100 thermo-couple by Thies Clima and a HMT337 humicap sensor by Vaisala,
respectively. Wind speed and direction at 10 meter height were obtained from a Combined Wind Sensor
Classic by Thies Clima (see Maturilli et al., 2013). At about 50 m distance, the radiation measurements





for the Baseline Surface Radiatio Network (BSRN) provide among others the downward solar radiation
detected by a Kipp&Zonen CMP22 pyranometer (Maturilli et al., 2015). The meteorological and surface
radiation measurements are available in a 1 minute time resolution via the PANGAEA data repository
(Maturilli et al., 2020). The daily/hourly mean values of the meteorological parameters were used in the
statistical analyses of the 80-days/3-days experiment and in Figure 2 and Figure 3 (the physico-chemical
parameters from the snow samples are instead punctual value).

**2.6 Statistical Analysis**

Multiple linear regression was carried out to evaluate the associations between the observed

surface snow rBC mass concentration and a set of predictors corresponding to the considered
meteorological and snow physico-chemical parameters. The regression models describe variation in rBC
concentrations as a function of atmospheric eBC concentration, surface snow coarse mode particles
number concentration, snow internal temperature (7 cm depth for 80-days experiment and 2 cm depth for
the 3-days experiment), snow precipitation, solar radiation and conductivity. The atmospheric
concentration of eBC was included in the model as a potentially proxy to explain the rBC mass
concentration in the surface snow. Other atmospheric parameters were initially considered, i.e. the wind
speed and direction, and the atmospheric stability (expressed as vertical wind speed); however, they were
removed because preliminary statistical analyses indicate that none of them is associated with the
observed variations in snow rBC mass concentrations. The number concentration of coarse mode particles
and the total electrical conductivity were included in the model in order to check common transport and
deposition pathways and similarities/differences as a response to the snow melting. Snow temperature and
the total incoming solar radiation were used to consider the thermodynamic processes occurring at the
snow surface, as melting or condensation (surface hoar).

Since the predictors considered in the linear regression models for the two experiments are

characterized by rather different measurement scales, results are reported in terms of standardized
estimated coefficients obtained by fitting the regression model after standardizing the variables. The
standardization simplifies the comparison among the different variables and between the two
experiments, in this way facilitating the data interpretation and discussion.

Further details about the statistical analyses are given in the Supplementary material.


**2.7 Back trajectories calculation and Potential Source Contribution Function analysis**

Air mass back-trajectories (BT) were calculated using the NOAA ARL HYSPLIT 4 rev. 513

transport model (Stein et al., 2015). Global Data Assimilation System (GDAS) meteorological input
fields with 0.5x0.5 degree resolution and a propagation time of 240 hours was employed. The trajectories





were calculated every hour for an endpoint of 500 m above ground level in Ny-Ålesund. A potential
source contribution function (PSCF) analysis has been applied to the BTs exploiting a specifically
developed FORTRAN computer code (Petroselli et al., 2018). That analysis considered BC concentration
measured in the air by both AE31 and PSAP. Briefly, the method calculates the probability of finding a
source of a particular pollutant on a certain region by superimposing grid cells to it and estimating the
fraction of the total time spent on each cell by trajectories associated with a high concentration measured
at the receptor site. The 90th percentile was used to define the high concentration limit and cells of 3 x 3
degrees (lat-long) were exploited in the calculation of probabilities. Details of the PSCF methodology
employed here are described in Petroselli et al., (2018). The data of the active fires, covering the last 12
days before the sampling day, are from the MODIS active fire products (https://firms.
modaps.eosdis.nasa.gov/firemap/), offered by NASA LANCE.

## 3. Results and Discussions

### 3.1 Seasonal snow surface rBC variation

Seasonal rBC snow surface concentration changes were investigated for approximately 80 days.
This experiment was performed in 2014 and covers approximately the entire spring periods until the snow
pack melting. The results of the atmospheric measurements and from the analyses of the surface snow
samples are reported in Figure2.

### 3.1.1 Atmospheric eBC concentrations

During the experiment period, the atmospheric eBC concentration show a remarkable variability
ranging from 80 ng m$^{-3}$ to < 5 ng m$^{-3}$. More in details, the highest concentrations were measured at the
beginning of the campaign, especially from April 15 to 27, followed by a general decreasing trend
characterized by the presence of several concentration peaks (on May 8, 17 and 24). Eurasian fires were
suggested as the main source of biomass burning tracers during spring 2014 (Feltracco et al., 2020).
In order to evaluate the impact of the Eurasian fires on the measured atmospheric eBC
concentrations, a thorough back-trajectories analysis was performed for both the snow-sampling periods.
Results of PSCF analysis on eBC (Figures SI 1a, SI 1b and SI 1c; open-fire episodes are reported in red
on the map) show a clear maximum of probability over the Central Siberia, which appears to be the major
source area of eBC in this period over Ny-Ålesund. Some false positive source areas are located in
Greenland, the Queen Elisabeth Islands region and the Arctic Ocean, even if associated to a lower
probability. These artifacts are due to the persistent circulation of BTs in the Arctic vortex. An example of
BTs generating the above salient features in the PSCF plot is reported in Figure SI 1b. Here BTs are
shown to loop for few days around the Arctic at high altitudes and afterwards to descend at lower


altitudes over Siberia, just four days before reaching Ny-Ålesund on April 22, when a clear maximum in
the eBC trend has been recorded. Back trajectory analysis supports the idea that the peaks of eBC in the
atmosphere in early spring are directly correlated with long-range transport from Eurasia, whereas the
peaks in late May and June are much lower in intensity, seemed to be more related to a Western
circulation pattern. The ammonia daily concentration time series (the only available biomass burning
tracer for that period in the area) measured at the Zeppelin station is overlapped to the Gruvebadet
atmospheric BC measurements in Figure SI 3. Biomass burning is known to be a significant source of
atmospheric ammonia (Andreae and Merlet, 2001). As shown in Figure SI 3, the two time series have a
similar behavior at the very beginning of the campaign, from April 3 to 8 and during the period between
May 7 and 21.

**3.1.2 Surface Snow/Atmospheric Aerosol Content and Atmospheric Conditions**
During the 80-days sampling period, wind was characterized by the following median values (25[th]
and 75[th] percentiles) of direction and speed:  205° (152°, 257°) and 2.7 (1.9, 3.7) m s$^{-1}$, respectively,
therefore coming mostly from South-West (Figure). Daily air temperature increased during the campaign
from -15°C to about +5°C (Figure 2), blue and red bars, represented as red bar in the legend), showing an
average value and standard deviation of -3.5 ± 5.8 °C. The temperature increase followed the seasonal
increase of the daily mean incoming solar radiation (Figure 2, orange line), increasing from
approximately 100 to 300 W m$^{-2}$, with an average of 185 ± 75 W m$^{-2}$. The snow precipitation episodes are
accounted for through the daily amount of deposited snow (Figure 2, blue bars), ranging from zero to 12
cm. The atmospheric eBC mass concentration, derived from the PSAP absorption coefficient, shows a
decreasing trend during the campaign and a remarkable variability, ranging approximately from 2 to 80
ng m$^{-3}$, with an average of 34 ± 23 ng m$^{-3}$.
The snow rBC mass concentration shows a significant variability over the 80 days, ranging
approximately from 0.2 to 6 ng g$^{-1}$ (Figure 2), with an average of 1.4 ± 1.3 ng g$^{-1}$, and it is in agreement
with results available in the literature (Mori et al., 2019; Jacobi et al., 2019; Aamaas et al., 2011). The
spatial variability of BC was assumed to be comparable to the results described in Spolaor et al. (2019)
for other particulate species in samples from the same field campaign. Nine samples were collected at the
same time in the designed sampling area and the results show a spatial variability in the order of 5 to 15%
(for sodium, mercury and iodine; Spolaor et al. (2019). An increasing trend can be observed for the rBC
mass concentration in the surface snow during the sampling period. By comparing the observed values of
this study with the results reported in Hadley and Kirchstetter (2012), we can estimate a very low snow
albedo reduction due to the BC particles in our samples, always lower than approximately 0.01 (raw
estimate). Given the low measured rBC mass concentrations, we decided not to calculate the BC radiative





impact. Moreover, this study lacks any detailed description of the snow physical conditions, like the grain
size, important for assessing realistic snow albedo reductions (Hadley and Kirchstetter, 2012; Skiles and
Painter, 2019). The median of the rBC mass equivalent diameter in the snow is $313 \pm 35$ nm (Figure),
similar to what obtained in other studies (e.g. Schwarz et al., 2013). The rBC mass equivalent diameter
show high variability, ranging from 200 to 500 nm (however, since the rBC concentrations were low the
evaluation of the particles geometric mean diameter for the biggest sizes, above 300/400 nm, has only to
be considered as qualitative given the high noise in the size distributions).

The number concentration of coarse mode particles (Figure 2, blue line) shows a constant

concentration in the first half of the campaign, until May 11, whereas increasing in the second half,
especially after the 1st of June, in concomitance with the onset of the snow melting period; the average
number concentration is $4914 \pm 4109$ # ml$^{-1}$. Also the conductivity (Figure 2, green line) shows an
increasing trend at the end of the sampling campaign when snow is melting, with an overall average value
of $30 \pm 8$ µS.

**3.1.2 Variables explaining the snow rBC mass concentration variability**


The fitted regression model for the 80-days experiment data explains 69% of the variance of

snow rBC mass concentration (R2=0.69) and indicates  a statistically significant association of the snow
rBC mass concentration with the coarse-mode particles number concentration ($p < 0.001$),  the amount of
snow precipitations ($p < 0.05$) and   the snow temperature ($p < 0.001$). The relations with the other
predictors are non-significant (see Table 1, reporting the standardized estimated coefficients and the
corresponding p-values). Figure 4 displays the 95% and 90% confidence intervals for the standardized
estimated coefficients. Intervals that do not include the zero correspond to statistically significant
predictors of the snow rBC mass concentration. If a confidence interval consists of positive values, then
there is a significant positive association between the corresponding predictor and snow rBC mass
concentration. Vice versa, if the confidence interval consists of negative values, then the association is
negative. Figure 4 displays both the confidence intervals for the 80-days campaign and the 3-days
experiment in a way to allow an immediate visual comparison of the estimated statistical associations
between the snow rBC mass concentration and the considered predictors.

In order to interpret the statistical results, the description of the 80-days campaign is split into two

periods depending on the temperature and the state of the snow. These periods identify the transition from
the "cold" to the "melting" state. The first period occurred before the end of May: in this period the rBC
mass concentration often increases in concomitance, or one day after, to most of the observed snowfall
episodes (April 9/10/11 and 17; May 17, 22 and 27/28; June 1), with exceptions for April 24 and May 7.
The sampling was performed in the late morning regardless the beginning/duration of the precipitation



events. Over the sampling period, a weakly statistically significant ($p < 0.05$, Table 1) positive relation was found between snow rBC mass concentration in surface snow and the daily amount of snow precipitation. Given the complexity of the system, the short sampling period and daily frequency and the intrinsic internal variability, this positive association can only be tentatively linked to the BC wet deposition process, removing 50% - 60% of the total atmospheric BC burden in the Arctic (Liu et al., 2011; Jacobi et al., 2019). In our study, the impacts of the wet deposition could be partially masked due to the sampling frequency and timing. However, our observations show that, on a daily scale, the precipitation episodes are not clearly related to a decrease in the atmospheric eBC mass concentration (Figure 2). Nonetheless, a doubtful negative association was found in the fitted statistical model between the atmospheric eBC and snow rBC mass concentrations, with an associated p-value of 0.061.

In the second period, from the beginning of May on, the atmospheric temperature increases and the surface snow starts melting, inducing post-depositional effects on the snow impurities content. At the beginning of June, the snow rBC mass concentration increases up to approximately 5 ng g$^{-1}$ and also the coarse mode particles number concentration increases remarkably (peaking between June 4 and 7). This positive relation is confirmed by the fitted regression model that indicates that the number of coarse mode particles is indeed the predictor with the highest significance level ($p < 0.001$, as reported in Table 1 and Figure 4). This positive relation can be explained by considering several processes affecting the BC and the coarse mode particles. Firstly, dry deposition is the main depositional process for the coarse mode particles, but recently it has been shown to be as well to provide a significant contribution in explaining the BC particles deposition (Liu et al., 2011; Jacobi et al., 2019). Secondly, it is possible that local sources become important contributors in the "melting" period due to the snow melting and the consequent exposure of soil and rocky areas in the surrounding of the sampling area. Ny-Ålesund was a mine town and the impact of local sources might have a non-negligible impact during the periods with little snow cover (autumns and late spring) and years with limited snow precipitations. Wind resuspension of BC (or other unknown refractory materials) and dust particles from uncovered areas eventually deposited on the remaining snow surfaces cause an acceleration of the melting and, as a consequence, a reduction of the snow season (positive feedback). The impacts of this positive feedback would be enhanced in a warmer climate where the activation of local sources would be longer in early winter and earlier in springtime.

Thirdly, we could explain the simultaneous increase of rBC mass and coarse mode particles number concentrations via post-depositional processes, as snow melting or sublimation, as visible between June 3 and 7-8. The episodes of snow surface melting can greatly affect the snow particulate content and we hypothesize that the hydrophobicity of pure BC particles, and of several species in the coarse mode particles, might affect its physical location in the snowpack during melting-refreezing





episodes (in the literature the response of the BC particles is still debated): the hydrophobicity of the
particles can cause the surface concentration to increase while losing water mass through percolation.
Moreover, during the sublimation episodes, the losses of surface water mass lead to an increase of the
particulate matter in the first layer of snow. The subsequent rBC mass and coarse mode particles number
concentrations decrease can be speculatively explained with the complex behavior of the snow mantle
during the strong melting and refreezing cycles and snow mantle collapsing. In conclusion, the processes
causing the similar behavior observed in this study are complex to disentangle and full closure
experiments are needed to tackle this subject, even though extremely complex and hardly manageable.

In this study, the estimated statistical association between snow rBC mass concentration and the

daily snow temperature is negative and strongly significant (p < 0.001, Table 1 and Figure 4). During the
80 days experiment we can distinguish two events where the temperature appeared to play a role in the
BC concentration, with an increase in rBC mass in the surface snow during the melting/refreezing
episodes (in agreement with the results of Aamaas et al. (2011). The first event occurred between May 5
to 12 and the second after May 20, when the proper snow melting began (Figure 2). The first event was
characterized by a rapid rise of daily air temperature (from -6°C to -1°C) in concomitance to a snow
precipitation event, followed by a rapid temperature decrease to -6 °C. The surface snow (10 cm)
mirrored this behavior first rising from -6 °C to 0°C and then cooling down to -6 °C. During this warm
event, the upper snow strata underwent a melting with likely surface water percolation, making the
surface BC concentration to increase. The second event started approximately on May 20 and lasted until
the end of the experiment (Figure 2). During this period, the atmospheric temperature increased
constantly, and the snow pack started to melt constantly. Moreover, surface BC concentration increased
almost continuously from May 25 to its maximum observed in June 6. Afterwards, the upper snow rBC
mass concentration tended to decrease following the rapid snow pack decline.

**3.2 Diurnal variation of rBC in surface snow**
**3.2.1 Surface Snow/Atmospheric Aerosol Content and Atmospheric Conditions**

The 3-days experiment was performed at the end of April 2015, during the Arctic spring. The

samples were collected on an hourly basis over 3 days achieving a high-resolution sampling frequency.
The wind direction and speed were quite constant (with median, 25[th] and 75[th] percentiles values of: 147°,
132°, 174° and 1.9, 1.4, 2.9 m s[-1], respectively) during the sampling period, blowing from South/Southeast
or Southwest. The atmospheric and surface snow temperatures remained well below 0°C (Figure 3), with
average values of -7.5 ± 1.2 °C and -8.3 ± 2.2 °C, respectively. The warmest air and snow temperature
were approximately -5°C, therefore excluding surface snow melting to happen.



The atmospheric concentration of eBC in ranged from 2 to 50 ng m$^{-3}$, decreasing during the

sampling period and not showing any particular diurnal pattern (Figure 3). The mean value of the

atmospheric eBC mass concentration is $34 \pm 23$ ng m$^{-3}$, similar to the average of the 80-days experiment.

The time series of the hourly eBC mass concentration is not showing any similar variability with snow

rBC time series, except for the common decreasing trend.

The surface snow rBC mass concentration exhibited hourly variability, showing up to 2-fold

hourly increases (especially during the first day), overlapped to a quasi-daily cycle (Figure 3, bottom

panel, smoothed dark blue line). rBC mass concentrations of approximately 15 ng g$^{-1}$ were measured from

the beginning of the sampling to the end of the second day, and of about 5 ng g$^{-1}$ from the beginning of

the third day till the end of experiment (Figure 3). The average value over the sampling period is $9.5 \pm 5.2$

514        ng g$^{-1}$ (approximately 6 times higher than that during the 80-days experiment). This higher BC

concentrations are probably due to the limited number of snow episodes during this year, compared to the

516        year of the 80-days experiment, causing a higher impact of the dry BC deposition (before the snow

event). As reported for the 80-days experiment results, by considering the results from Hadley and

Kirchstetter (2012) it is possible to estimate a low snow albedo reduction of approximately 0.02-0.03

(difference between the albedo of pure snow and the albedo of snow with BC particles). Given the low

measured rBC mass concentrations, and the lack of detailed snow grain size measurements, we decided

not to calculate the BC radiative impact. Moreover, this study lacks any detailed description of the snow

physical conditions, as the grain size, important to assess realistic snow albedo reductions (Hadley and

Kirchstetter, 2012; Skiles and Painter, 2019). The rBC mass size distribution, instead, was characterized

by a median value of the geometric means of about $230 \pm 32$ nm, significantly lower than that which was

measured during the 80-days, and still in agreement with previous studies (Sinha et al., 2018; Bond et al.,

2013). The concentrations of EC and OC measured in parallel snow samples (not of the same volume) are

as well reported and described in Figure SI 4; the interpretation of the differences between the rBC and

the EC measurements in snow samples is beyond the objectives of this manuscript.

The number concentration of coarse mode particles remains virtually stable in the first half of the

campaign, until the end of April, and shows an average value over the whole three days of $26642 \pm 9261$

# ml$^{-3}$ (approximately 5.5 times higher than during the 80-days experiment). The water conductivity

shows a similar behavior, and it is characterized by an average of $39 \pm 9$ µS (30% higher than during the

80-days experiment).

All the measured snow impurities time series show two common features: first, a decrease in

absolute values was detected between 4 and 8 a.m. of April 30, despite the absence of precipitations and

of any particular meteorological episode (Figure 3); second, the impact of the snow precipitation events

from approximately 4 p.m. to midnight of the April 30, where the concentrations of aerosols in the snow



slightly increased at the very beginning whereas decreasing at the end of the event. Only the BC core
diameter remained above the average when the other aerosol content decreased (up to approximately 400
nm of mass equivalent diameter), consequently going back to the average value. Also in this experiment,
the spatial variability was estimated to account overall for the 5-15% to the total variability of the
measured parameters (Spolaor et al., 2019).

The results of BT analysis for the 3-days experiment are reported in Figure SI 2, suggesting that

the air masses were persistently circulating in the polar vortex and very similar within the three days in
terms of BC atmospheric sources, physical properties and mixing state. The daily average concentration
of ammonia, measured at the Zeppelin observatory, are similar to the lowest values measured during the
80-days experiment (approximately 0.5 $\mu$g L$^{-1}$), suggesting a background regime with a two-times
increase during the last day (Figure SI 4). This result suggests what observed with the BT analyses:
similar air masses during the sampling period with a low/negligible impact of the Northern Siberian fires
(quite low in number during those days, Figure SI 2).

**3.2.2 Variables explaining the snow rBC mass variability**

The results obtained from the 3-days experiment have been evaluated using the same statistical

approach considered to explain the rBC snow mass concentration variability in the 80-days experiment.
The statistical model for the 3-days experiment explains 83% of the total snow rBC mass concentration
variance, a percentage higher than the 80-days experiment, likely due to the more stable conditions. The
fitted model indicates a statistically significant association between the rBC mass concentration in the
snow and the conductivity ($p < 0.001$, Table 1), the number concentration of coarse-mode particles ($p <$
$0.01$, Table 1), the snow precipitation amount ($p < 0.001$, Table 1), the incoming solar radiation ($p < 0.01$,
Table 1) and the snow temperature ($p < 0.05$, Table 1). The standardized estimated coefficients are
reported in Table 1, displayed along with 90% and 95% confidence intervals in Figure 4.

The positive statistical association of total snow rBC mass concentration with the conductivity of

the melted snow samples ($p < 0.001$, Table 1 and Figure 4) can be explained by a simultaneous deposition
of different aerosol species (as sea salt with oceanic sources). For instance, air masses following pathways
over the ocean after having entrained BC particles from biomass burning episodes will result in a positive
correlation of snow BC and conductivity.

The association between the coarse-mode particles number concentration and the snow rBC mass

concentration is positive and strongly significant ($p < 0.001$, Table 1 and Figure 4), similarly to what was
observed for the 80-days experiment results.

A negative association is found between the rBC mass concentration in the snow and the

incoming solar radiation ($p < 0.01$, Table 1 and Figure 4), and a weaker negative association with the



snow temperature (p < 0.05, Table 1 and Figure 4), with the latter being strongly dependent on the solar
radiation. This relation confirms what observed in Figure 3: the rBC mass concentration in surface snow
follows a diurnal cycle, lower when the solar radiation is higher and vice versa. The quasi-daily cycle of
rBC mass concentration in the surface snow layer has never been studied and reported in the literature.
The BC particles are known to be non-volatile and non-photochemical active, therefore the decrease in its
concentration observed when the solar radiation is higher could not be explained as a re-emission process
from the snow back into the atmosphere as observed for other aerosol species (Spolaor et al., 2018;
Spolaor et al., 2019). The results show that the highest rBC mass concentration levels are detected in
samples collected in the late afternoon. The late night/early morning concentration decrease is connected
with surface hoar formation able to dilute the surface snow BC concentration.  Specifically, the lowest
rBC mass concentration values is found between 5 and 12 am and in the same time interval the solar
radiation increases from 100 to 400 W m$^{-2}$, followed with a delay by air and snow temperatures increase.
In these periods, the temperature offset between the air and the surface snow is the highest, up to 4°C,
with the surface snow being the coldest between the two. Condensation of water vapor on the top of the
snow crystals is possible, adding snow mass without BC in the collected samples and diluting the original
rBC mass concentration. This process may lead to an overall negative feedback on the BC radiative
impact, making its concentration to decrease during the daily maximum of solar radiation.
The snow precipitation amount is negatively associated with the rBC mass concentration in the
snow (p < 0.001, Table 1 and Figure 4). As previously remarked, the aerosol scavenging intensity is not
measurable with snow sampling strategies based on the sampling of a constant snow thickness from the
surface (3 cm in this case). The negative relation observed in this study is due to the high frequency
sampling, being able to follow the evolution of the BC particles scavenged during a snow episode (from 3
to 12 p.m. of the 30[th] April 2015). The beginning of the precipitation episodes appeared to remove the
highest amount of BC particles, leaving the atmosphere cleaner as reflected by the lower BC mass
concentration revealed in subsequent samples. The snow collected at 18:00 of April 30 showed a higher
amount of rBC as well as the highest coarse mode particles number concentration and conductivity. In the
next few hours, from 9 to 12 p.m., the snow precipitations were very "clean" in terms of aerosol content
and rBC mass concentration.
From the 3-days experiment, it appeared that the snow surface physical processes like surface
hoar formation and sublimation play an important role, and that the physical characteristics of the snow
layers in which BC is embedded should be more studied in order to better characterize the daily variations
of BC and its impact on the albedo. The 3 days experiment took place under clear sky conditions (most of
the time) and this is of high importance for the variations observed. Indeed, surface hoar can only form
under clear sky when the snow surface is cooler than the air due to longwave radiation emitted and lost,





and under calm weather with low wind. Under other conditions, cloudy weather for example, the BC
diurnal variation may show a completely different pattern, as snow will likely be affected by longwave
radiation backscattered by clouds toward the snow surface, and melting and/or sublimation at the snow
surface will only be observed, but likely no condensation of atmospheric water vapor. Furthermore, these
daily variations showed that the highest concentration of rBC is measured during mid-day/afternoon,
when the incoming radiation amount is still high, and that may significantly affect the amount of extra
energy absorbed by the surface snow, further enhancing metamorphism and feedback processes. More
detailed studies including snow density and optical snow grain radius measurements should be pursued
and at a cm vertical resolution in order to correctly estimate the radiative impact of the daily rBC
variations.
**4. Conclusions and Future Perspectives**
The two experiments suggested that the main drivers of the rBC mass concentration variation in
the Svalbard surface snow are mainly precipitations events, snow metamorphism (melting and surface
hoar formation and sublimation), and potentially the activation of local sources during the melting periods
triggering a positive-feedback based on the snow albedo reduction. On a daily frequency (80-days
experiment) coarse-mode particles are associated to the snow rBC mass concentration, even in periods
characterized by the influence of biomass burning emissions. On an hourly frequency (3-days experiment)
the snow deposition and the daily solar radiation cycle appeared to be mostly controlling the surface snow
rBC content under clear sky, via hoar formation/condensation processes, with the coarse mode particles
number concentration positively associated with it. The absolute rBC mass concentration resulting in a
minor or negligible snow albedo reduction of approximately the 3% at maximum (see Hadley and
Kirchstetter, 2012).
During the seasonal time scale (daily sampling strategy), the multilinear statistical model was
able to explain 69% of the surface snow rBC mass concentration variance. Our results indicate a positive
association between the snow rBC mass concentration and the coarse-mode particles number
concentration, due to similar responses to dry and wet deposition processes and comparable behaviors in
the presence of post-depositional processes. The amount of rBC in the surface snow appeared to be
statistically de-coupled from the eBC atmospheric load. The importance of the wet-deposition process
was statistically highlighted in both experiments.
Long-range transport and melting-induced activation of local sources are key parameters in
describing the BC origin in the atmosphere and in the surface snow in the Ny-Ålesund area (and might in
a large portion of the Svalbard archipelago), acting with different intensities during the year. However,
our results suggest that despite possible high atmospheric BC concentrations as in the case of long-range



transport of biomass burning plumes, the surface snow rBC mass concentration can be almost completely
unaffected in the absence of snow precipitation events. During the surface snow melting period (with
atmospheric temperatures above 0°C) we revealed an increase of snow rBC mass and coarse-mode
particles number concentrations, suggesting an increase of the impact of the local sources, activated by
the snow melting leaving the surface exposed to winds.  Moreover, this measured increase in the snow
aerosol load is influenced by the response to melting-refreezing cycles and water mass loss. Both these
mechanisms cause an increase in the snow surface insoluble particles concentration, causing a positive-
feedback mechanism enhancing their radiative impacts and fastening the snow melting.
83% of the hourly/daily variance was explained by the statistical model, again resulting in a
positive association with the coarse-mode particles number concentration, suggesting similar responses to
the depositional patterns and responses to post depositional processes (condensation of water vapor on the
top of the snow crystals). The negative association with the solar radiation and the temperature of the
snow suggest that part of the rBC mass variability in the snow undergoes to a daily cycle linked to snow
metamorphism processes, as sublimation and condensation of atmospheric water vapor. The condensation
of water vapor on the upper layer of the snow makes the rBC mass concentration decrease by dilution
(when keeping a constant sampling thickness), especially during the hours of the day when the solar
radiation is at its maximum: however, the complex snow crystals-radiation interaction makes it difficult to
evaluate the radiative impact of this process.

**Acknowledgements**
This work was part of the PhD (in "Science and Management of Climate Change") of Michele Bertò at
the Ca' Foscari University of Venice that was partly funded with the Early Human Impact ERC project.
Thanks to Giuseppe Pellegrino for helping collecting the samples. Thanks to Jacopo Gabrieli and the
technicians of the Ca'Foscari University of Venice for the precious help in building up the coarse mode
particles and conductivity measurement apparatus. We acknowledge the use of data and imagery from
LANCE FIRMS operated by the NASA/GSFC/Earth Science Data and Information System (ESDIS) with
funding provided by NASA/HQ. We want to thank Paolo Laj and the LGGE (Grenoble, France) for
lending us the SP2 and Marco Zanatta for transferring the SP2 know-how on instrumental functioning and
data analyses. Thanks to Martin Gysel-Beer, PSI, for the IGOR based SP2 Toolkit for SP2 data analyses.
We thank Marion Maturilli and AWI for providing us with the meteorological data. Thanks to Giorgio
Bertò for checking and correcting the language of this manuscript. This paper is an output of the AMIS
project in the framework of "Project MIUR – Dipartimenti di Eccellenza 2018-2022". This project has
received funding from the European Union's Horizon 2020 research and innovation programme under





grant agreement No 689443 via project iCUPE (Integrative and Comprehensive Understanding on Polar
Environments).

**Data Availability**
Meterological and surface radiation data are available at the PANGAEA database (Maturilli, 2015a;
2015b; 2015c; 2016a; 2016b; 2018a; 2018b; 2018c; 2018d; 2018e). The data for precipitation amount at
Ny-Ålesund can be accessed via the eKlima database of MET Norway. The BC data are available upon
request.

**Author Contributions**
Author contributions. AS, EB, DC and MB conceived the experiments; AS, EB, DC, and LP collected the
samples; MB measured the samples; KM and MMaz provided the atmospheric eBC concentrations; SC
and DC provided the back-trajectories analyses; CV performed the statistical analyses with inputs from
MB and AS. MB prepared the manuscript mainly with inputs from AS, J-C. G and DC (in the methods
section from AS, KM, MMaz) and all co-authors contributed to the interpretation of the results as well as
manuscript review and editing.

**Data repository**
Maturilli, Marion (2020): Basic and other measurements of radiation and continuous meteorological
observations at station Ny-Ålesund  (April, May 2014 and April, May, June 2015), reference list of 10
datasets.     Alfred     Wegener     Institute     -     Research     Unit     Potsdam,     PANGAEA,
https://doi.pangaea.de/10.1594/PANGAEA.913988 (DOI registration in progress)










**FIGURES**
**Figure 1.** a) Experimental sampling site location (dark grey rectangle), in proximity of the Gruvebadet
Aerosol Laboratory. b) Gruvebadet area (black square), close to the Ny-Ålesund research village. From:
Spolaor et al., 2019 (maps from https://toposvalbard.npolar.no/)

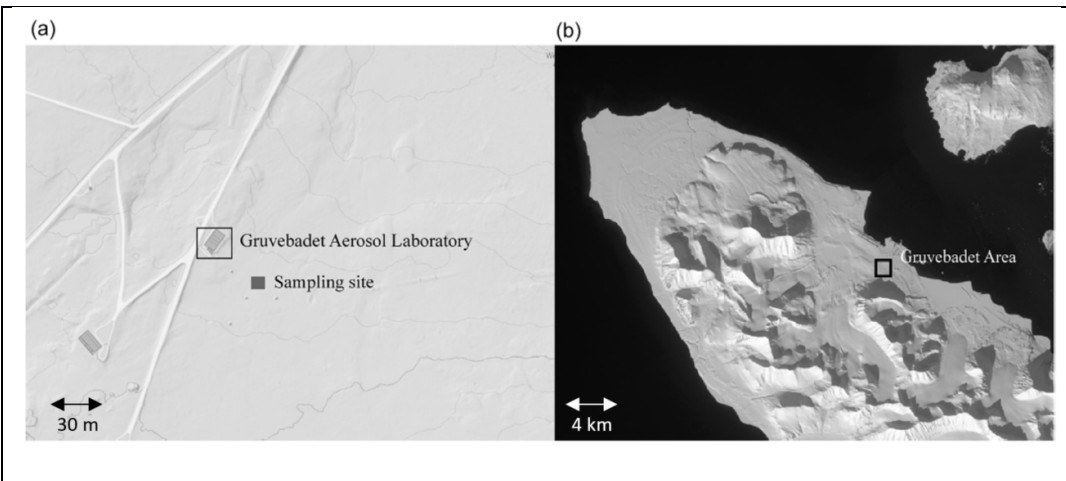

















**Figure 2.** The 80-days experiments daily snow samples rBC mass concentration (light blue), eBC mass
concentration in the atmosphere (black), geometric mean mass equivalent diameter (purple), number of
coarse mode particles (blue), total conductivity (green), meteo/snow parameters used in the statistical
exercise: wind speed color coded for wind direction, solar radiation (orange line), air and surface snow
temperatures (blue bars and green line respectively), amount of fresh snow ("snow precipitations", light
blue bars) and the snow accumulation ("Neg. accumulation"; the values where multiplied by -1 in order to
show the similar trend of the snow lost and of the air/snow temperature during the melting period at the
end of the campaign).

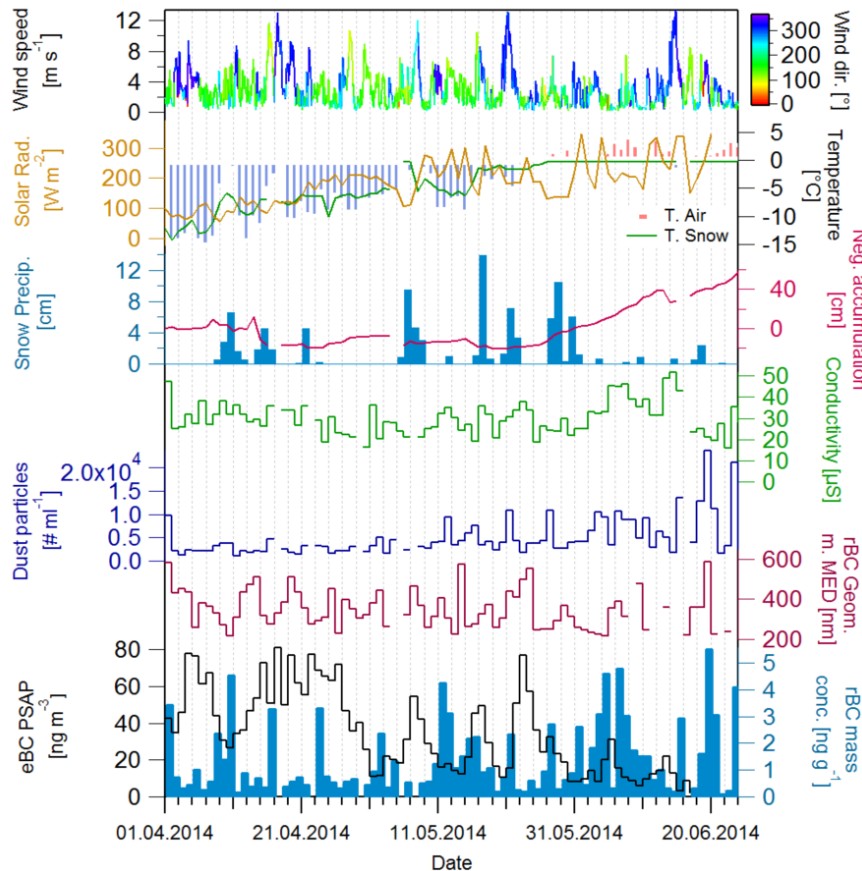






**Figure 3.** The 3-days experiments snow samples hourly rBC mass concentration and smoothed line (light
blue bars), atmospheric eBC mass concentration in the atmosphere (black), geometric mean mass
equivalent diameter (purple), the number concentration of coarse mode particles (blue) and the total
conductivity (green), meteo/snow parameters used in the statistical exercise: wind speed color coded for
wind direction, solar radiation (Orange line), Air and surface snow temperature (blue bars and green line
respectively), amount of fresh snow ("snow precipitations", light blue bars). The yellow bars are centered
on the midnight hours for the three sampling days.

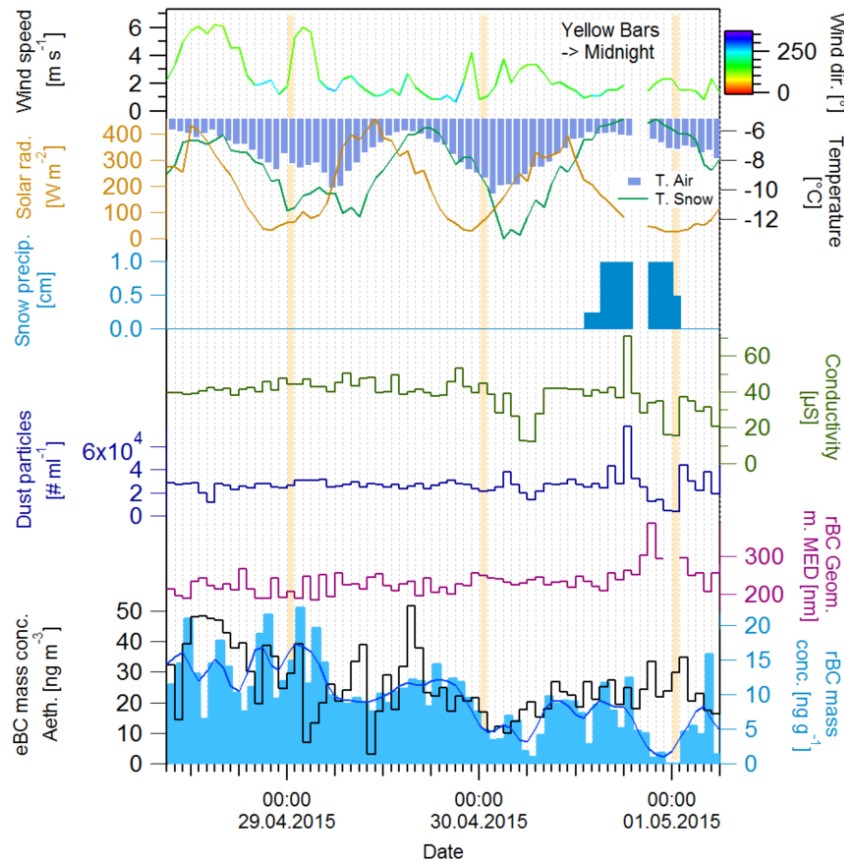







**Figure 4.** Standardized estimated coefficients of the multiple regression models fitted to the 3 days and 80 days experiments. The segments correspond to 95% confidence intervals about the corresponding estimates. The internal thicker segments correspond to 90% confidence intervals. The abbreviations used in the plot are: "log(cond)" – logarithm of the water conductivity time series, "log(dust)" – logarithm of the coarse mode particles number concentration time series, "eBC" – equivalent black carbon atmospheric concentration, "snow" – amount of fresh snow from the precipitation episodes, "SWR" – solar radiation, "temp" – the snow temperature.

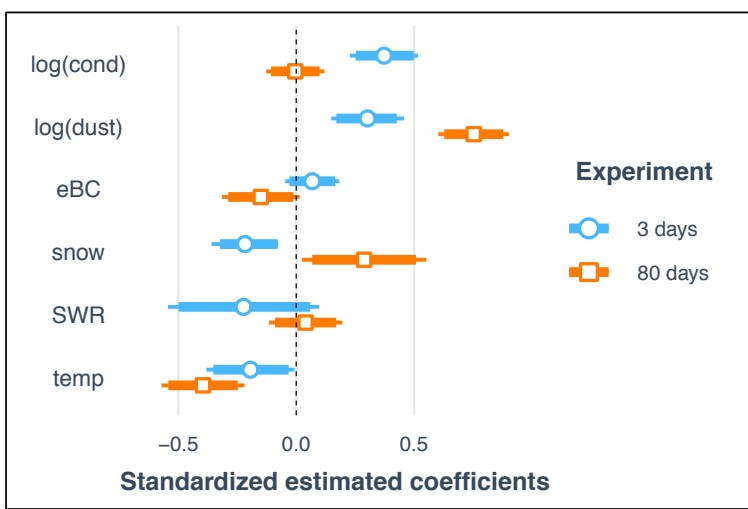



**TABLES**

**Table 1.** Standardized estimated coefficients of the regression models for the 3 days and 80 days experiments. Standard errors are reported within parentheses below the corresponding estimate. The intercept and the trigonometric terms are not displayed.

| Predictor | 3 days | 80 days |
|---|---|---|
| **log(cond)** | 0.38 *** | -0.00 |
| | (0.07) | (0.06) |
| **log(dust)** | 0.23 ** | 0.75 *** |
| | (0.07) | (0.08) |
| **eBC** | 0.06 | -0.15 |
| | (0.05) | (0.08) |
| **snow** | -1.02 *** | 0.29 * |
| | (0.19) | (0.13) |
| **SWR** | -0.43 ** | 0.04 |
| | (0.16) | (0.08) |
| **temp** | -0.23 * | -0.40 *** |
| | (0.09) | (0.09) |
| **$R^2$** | 0.83 | 0.69 |

*** $p < 0.001$; ** $p < 0.01$; * $p < 0.05$



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
