# Peer review of "Black Carbon Seasonal and Diurnal Variation in surface snow in Svalbard and its"

_Atmospheric Chemistry and Physics, 2020_

## Referee Comment (RC1) · Anonymous Referee #1 · 4 Aug 2020

This study aims at linking a number of observed variables with the concentration of rBC in surface snow. The purpose is to investigate processes regulating rBC mass concentration in snow by using a multi-linear model to explain the observed variability of rBC in surface snow.

In general, the manuscript show some signs of being rushed to submission, yet it would benefit from substantial trimming. Specifically the introduction could be shorter, the speculations could be fewer, and material related to trajectory calculations do not forward the scientific reasoning and can be omitted from the study.

The opening statement in the conclusions illustrates some of the problems mentioned

above "....the main drivers of the rBC mass concentration variation in the Svalbard surface snow are mainly precipitation events, snow metamorphosis (melting and surface hoar formation and sublimation), and potentially the activation of local sources during the melting periods triggering a positive-feedback based on the snow albedo reduction".

The first section could very well have been stated before the study and the proposed potential activation mechanism (assuming that "local sources" refers to dust) is never associated with any observed albedo reduction, nor snow temperature increase.

In its current form I cannot recommend publication of this manuscript and it requires substantial revision before it can be considered again. My main reason is the method used, which is the linear model based on essentially instantaneous observations, which may or may not be relevant.

In a column of snow there are not so many ways to change the concentration of BC. Assuming that BC is not removed from a column of seasonal snow (other during catastrophic melting events or special cases), then BC can only be added whereas water can be both added and removed. Over a season, the concentration of BC is not very much determined by the momentary environmental parameters, but rather the integrated deposition of BC (wet and dry) and the integrated flux of water. Hence, what shows up in late spring is the integrated contribution of the processes throughout the season. To get closer to a useful model, the time series could be transformed into cumulative variables where snow events are treated separately (as it covers up old snow and in a way puts some processes on hold).

My point is that the variables and method is not appropriate for the scientific question. On a daily frequency (even more so on an hourly basis) there is no real reason why the atmospheric concentration and the concentration in 10 cm (or even 3 cm) of snow should be related. The exception is during a precipitation event (cf. Hegg et al., 2011). Dry deposition is too slow and the volume of air above is too large for any observable

decisive impact on daily basis. For instance; to change the average concentration by 1 ng/g in 10 cm of snow at a density of 0.5 kg/L, an atmospheric concentration of 50 ng/m^3 and a Vd=0.3 mm/s (see line 125 in manuscript) will take about 2 months. The fact that dry deposition contributes to 100% of deposition between snow events does not make it very important on a seasonal time scale except in very arid regions (cf. Wang et al., 2014).

In spring, a round figure for "non-new" snow density could be 0.5kg/L, this makes 10 cm represent approximately 50 mm melted equivalent water. This number should be compared to the annual precipitation of approximately 380 mm. In other words, each daily sample of snow represents the same order of time as the whole 80-day campaign (give or take).

Based on this comparison above, one would not expect a large daily variation from samples taken almost side-by-side. However, my interpretation of the data is that there is indeed a strong signal of this variability in the data. See for instance Svensson et al. (2013). The authors reference Spolaor et al. (2019) and claim comparable spatial variability of 5-15%. But, the species observed in Spolaor et al. (2019) have very different chemical and physical properties compared to BC.

I agree with the authors that minerals and BC are likely to have similar properties, which eventually enhance their surface concentrations. This in combination with the spatial variability give these two variables the strongest link of the parameters tested in the linear model. The process most likely responsible for generating the observed spatial variability is snow drift (this process is not discussed by the authors). Sastrugies (that often form on the surface of snow from wind) are in the current context a travel in time and a process to generate small-scale variability of many snow impurities.

Instead of SW radiation and snow temperature (not entirely independent, as pointed out by the authors), it would be more appropriate to use the difference in water vapour pressure between the surface snow and the air above. This is readily estimated from

the snow temperature (outgoing LW can also be used) and RH and air temperature. This will explain the effect of hoar and evaporation on changes in BC concentration. These processes have both a seasonal and diurnal cycle, but the order of magnitude is typically less than 1 mm equivalent melt water per day. Again, this value should be compared to the amount of snow sampled each day in the two campaigns. Also, working with the difference in water vapour pressure will take care of the apparent time lag noted in Figure 3 by the authors.

In spring, most processes drive the concentration of BC to increase. The big exception is snow events, which will, with very few exceptions, reduce the surface concentration of BC where it fell. The fact that the authors appear to see an increase of BC especially just after a snow event could be an effect from the tractors clearing snow from the roads nearby the sampling site (cf. Figure 1a in manuscript and https://www.youtube.com/watch?v=2Y1YZONbfFY for a perspective).

There are minor questions that can be asked on details, but I feel there is a need to substantially revise the manuscript with a more clear idea about the relation between temporal and special scales and what processes (variable links) that are plausible in regulating rBC in surface snow. Perhaps my order of magnitude estimates above are wrong, but I encourage the authors to do similar assessments, nevertheless.

Hegg, A.D.,et al. Measurements of black carbon aerosol washout ratio on Svalbard. Tellus B, 63, 891-900, 2011. Svensson J. et al Observed metre scale horizontal variability of elemental carbon in surface snow 2013 Environ. Res. Lett. 8. Wang, ZW., et al., Elemental carbon in snow at Changbai Mountain, northeastern China: concentrations, scavenging ratios, and dry deposition velocities. Atmos. Chem. Phys. 14, 629-640, 2014.

---

## Referee Comment (RC2) · Anonymous Referee #2 · 11 Aug 2020

The goal of the presented analysis is to determine what factors drive changes in the concentration of BC in snow at a site in Svalbard. Snow samples were analyzed for rBC, conductivity (ions), and dust (coarse particle number count); atmospheric eBC was measured with a PSAP and/or aethalometer; and a range of meteorological variables (wind speed, wind direction, solar radiation, temperature, precip) were monitored. Concentrations of rBC in snow were correlated with these other variables, nominally to elucidate the cause of changes in snow concentrations. This was done for two periods: an "80 days" period, where snow was sampled daily and a "3 days" period, where snow was sampled hourly. For the "80 days" campaign, snow was sampled from the top 10cm of the snowpack. For the 3 days of hourly sampling, snow was collected from

the surface to 3cm depth.

The study suffers from a lack of analytical focus and robust conclusions, in large part stemming from the fact that it appears to take a bunch of variables and see what emerges, rather than starting with a hypothesis, then designing an experiment based on the hypothesis.

Fundamentally, the amount of BC in snow is determined by: atmospheric concentrations immediately above the snow surface and the dry deposition rate (dry deposition); atmospheric concentrations at and below cloud level, the wet scavenging rate of these aerosols, and precipitation amount; and post-depositional processes such as the addition of snow water without BC (e.g. hoar frost), loss of snowpack water through sublimation and melting, and the redistribution of BC in the snowpack with melting. If the goal was to determine what factors control the concentrations of BC in snow, the experiment should have been designed to quantify how these processes specifically affect BC in snow.

The approach of doing systematic sampling of the top 10cm or 3cm depth rather than over, e.g., distinct layers in the snowpack affected by different processes, confounds the ability to separate the role of different drivers. Changes due to dry deposition and hoar frost deposition would be best determined by sampling a very thin surface layer; changes due to deposition with new snowfall would be best determined by sampling the newly fallen snow and the previously snow layer separately; and changes due to the impacts of snow melt would be best determined by sampling multiple layers, with distinct samples for the layer affected by melt and then in layers below this. No reason is given for the selection of the 10cm and 3cm depth snow samples.

It's also difficult to understand why the suite of variables measured was selected. Why would changes in solar radiative flux alter the snowpack BC? Why measure the conductivity of the snowpack (ions)?

There is also a lack of clarity in the presentation regarding what variables could actually

drive change in snow BC, versus simply covary with them. Section 3.1.2 is titled "Variables explaining the snow rBC mass concentration variability, and therein it's stated that (lines 451-452) after snowmelt starts "the number of coarse mode particles is ... the predictor with the highest significance level." But this is not because the changes in dust concentration are actually driving changes in BC. One can only assume, as the authors do, that the two must be co-deposited, possibly from both being lofted from the nearby ground surface.

It's also not at all clear why the authors chose specifically to look at the diurnal cycle in the concentration of BC. (The authors assert that BC concentrations show a "quasi-daily cycle" but I really don't see this. The blue line pointed to in Figure 3 looks to me like it could just be smoothed random variations.) Other than the effects of hoar frost, which might deposit during one part of the day and perhaps sublimate during another part of the day, there isn't any reason to *expect* there to be a diurnal cycle in snow BC concentrations. The authors therefore attribute the diurnal cycle they claim to see in rBC concentrations to this process, but it's again rather hand-waving.

In the end, the factors that are most clearly seen to affect snow BC concentrations are things that we already knew a priori to be important: deposition with new snowfall and snow water loss in sublimation and melting. To this is added the resuspension of local sources of rBC during snow melt, though this is more of a theory than a robust finding. The study doesn't seem to provide any new quantitative information that would, e.g., be useful to improving modeling of processes driving snow BC concentrations. Further, it's not at all clear how generalizable the results of the study at this location are, especially in terms of the role of resuspension and hoar frost.

The finding that there isn't a correlation between the measured atmospheric BC and snow BC concentrations is not at all surprising; in fact, these two would only be correlated if dry deposition was the primary driver of BC deposition, and if the snow samples collected were of a sufficiently thin surface layer. The authors themselves note that ∼60% of the BC deposition at Svalbard is through wet deposition – and of course,

every time it snows, the previous surface layer is buried, confounding detection of the role of dry deposition through sampling of surface snow down to a fixed depth.

Beyond these issues the paper could be considerably shortened. It starts with a fairly broad overview of climate changes in the Arctic and previous measurements of BC in the Arctic. (Notably, the latter doesn't include one of the larger surveys of BC in Arctic snow that appeared in this same journal and that also included sampling from Svalbard: Doherty et al., 2010, "Light-absorbing impurities in Arctic snow"). The goal of the analysis was to reveal the causes behind *variations* of BC in snow; the absolute amounts and the radiative forcing are not the focus so this review of concentrations across the Arctic doesn't seem very relevant. What would have been more useful is a review of what other analyses to date have show about the processes that dominate variations in the concentrations of snow BC.

There is also extensive discussion of meteorological variables (e.g. winds) and back-trajectories really don't add anything to the analysis. These could be cut.

Overall, the paper would need to be significantly revised to be suitable for publication.